# A Novel Listwise Alignment Approach for Language Models with Explicit Rewards

## Abstract

Existing alignment techniques, including Direct Preference Optimization (DPO), are primarily designed for pairwise preference data where rewards are inferred rather than explicitly provided. In this paper, we propose a comprehensive framework for aligning large language models (LLMs) by introducing a new optimization objective that facilitates the processing of reward datasets, which consist of a list of responses explicitly marked with scalar preference scores. Our contribution includes the development of a novel algorithm, termed Soft Preference Optimization (LPO), which allows for the direct derivation of an LLM policy from both reward and preference datasets. At the heart of LPO is a unique listwise preference optimization objective formulated using an exponential-logarithmic function and an adaptive loss coefficient, which effectively integrates listwise preference signals into the LLM. We assess the efficacy of our approach under both reward and preference scenarios using different sizes of Mistral models. Experimental results indicate that our method outperforms several preference-based benchmarks, particularly when reward datasets are utilized. Additionally, our method demonstrates a significant advantage over DPO in intricate reasoning tasks, such as mathematical problem-solving and coding.

## 1 Introduction

Aligning pretrained Language Models (LMs) with scalar rewards that encapsulate human intentions is essential for improving their compliance with given instructions (Ouyang et al., 2022; OpenAI, 2023). Such rewards can be imparted either explicitly or implicitly. Explicit rewards include scalar evaluations from human assessors or sophisticated models such as GPT-4 (OpenAI, 2023), whereas implicit rewards come in the form of preference judgments made between pairs of responses. An effective method for aligning LMs using preference data is Direct Preference Optimization (DPO) (Rafailov et al., 2024b). DPO introduces a reward-training loss by parameterizing the reward model as the likelihood ratio of responses generated by two LMs, thereby enabling simultaneous training of the reward model and extraction of the LM's policy. This methodology is more efficient than conventional Reinforcement Learning (RL) techniques (Ouyang et al., 2022), which generally necessitate a two-step procedure: initial training of the reward model followed by the derivation of the LM's policy.

Despite its simplicity and effectiveness, DPO is designed specifically for preference data ($x \rightarrow \{y_w > y_l\}$) with $K = 2$ responses per instruction $x$. When faced with multiple responses, directly assigning a scalar reward to each response is generally more convenient and efficient than conducting pairwise comparisons. However, the resultant reward datasets ($x \rightarrow \{(y_i, r_i)\}_{1:K}$) cannot be directly utilized for DPO training. Prior studies (Tunstall et al., 2023) typically address this by pruning the reward dataset, selecting the top response and then pairing it randomly with another response. This approach is suboptimal since it discards both the reward values and additional non-preferred responses during the data preprocessing phase. Furthermore, a commonly observed issue with DPO is the tendency for the likelihood of the preferred response to diminish over the course of training (Pal et al., 2024; Rafailov et al., 2024a). Our analysis indicates that this phenomenon primarily stems from DPO's emphasis on adjusting the relative likelihood among various responses for each given instruction.

To tackle these challenges, we introduce Listwise Preference Optimization (LPO), an alignment technique that facilitates the direct extraction of language model (LM) policies from both reward and preference datasets, accommodating any number of responses. Notably, LPO encompasses the Direct Preference Optimization (DPO) loss as a special instance within the context of pairwise preferences, thereby positioning itself as a natural progression of DPO. While DPO operates as a single-pair ranking loss, LPO extends this concept to a listwise framework. Unlike DPO, which relies on the assumptions of the Bradley-Terry model, LPO is grounded in the principles outlined by Burges et al. (2006). Additionally, we introduce LPO-abs, an alternative approach to LPO, which incorporates a regularization term aimed at addressing the diminishing reward problem. LPO-abs diverges from LPO solely in its loss formulation but remains applicable to both preference and reward datasets.

We assess our methodology using the Mistral-7B models from two perspectives. When reward datasets (Cui et al.) are accessible, we demonstrate that directly employing our reward-based alignment yields a significant enhancement over preference-based algorithms, achieving superior evaluation scores when judged by the GPT-4 model. This improvement can be attributed to LPO/LPO-abs's capability to effectively utilize additional sub-optimal responses. In scenarios where only preference data is available (Yuan et al., 2024b), we benchmark the LPO-abs method against the DPO loss. Our comprehensive experimental results across multiple benchmarks indicate that LPO-abs surpasses DPO in handling intricate reasoning tasks, including mathematical problems and coding challenges.

Our main contributions are summarized as follows:

- We introduce LPO, a preference optimization method grounded in reward datasets. Additionally, LPO-abs addresses the data likelihood decline issue inherent in DPO. These two proposed methods are particularly well-suited for both reward and preference data, providing a comprehensive framework that integrates preference-based algorithms.
- We perform extensive experiments using both reward and preference datasets, illustrating that our methods significantly outperform various preference-based approaches by fully leveraging the information contained in reward datasets.

## 2 RELATED WORKS

**Development of language models.** Recently, the development of language models has undergone significant changes (Zhao et al., 2023), evolving from initial rule-based approaches to today's data-driven deep learning models. A milestone advancement began in 2018 when Google introduced the BERT model (Devlin et al., 2019) with a Transformer architecture (Vaswani et al., 2017), which achieved remarkable results in various natural language processing tasks through unsupervised training. Following this, even larger models like GPT-3 (Brown et al., 2020), ChatGPT (Ouyang et al., 2022) and GPT-4 (OpenAI, 2023) further propelled this trend, not only reaching unprecedented scales with hundreds of billions of parameters but also demonstrating versatility and flexibility across a diverse range of NLP tasks. These advancements have spurred researchers to explore applications of LLMs in areas such as machine translation, text summarization, dialogue systems, and even multimodal understanding.

While it is important to develop large language models in the size of 100 billions, the even wider applications rely on the relatively smaller sized models. These compact models, while perhaps not matching the sheer breadth of knowledge or depth of understanding afforded by their gargantuan counterparts (Zeng, 2023), offer significant advantages in terms of deployment flexibility, computational efficiency, and accessibility across diverse platforms and devices, including those with limited resources. Consequently, efforts aimed at refining smaller models to optimize both efficacy and practicality are critical for broadening the reach and utility of NLP solutions in everyday contexts. Recently, the community has witnessed the releases of open-sourced models like LlaMA-2 7B (Touvron et al., 2023), Mistral-7B (Jiang et al., 2023). These models can run an edge device like laptop or cellphone, after being deployed under frameworks like Ollama[1].

**Alignment of large language model.** The alignment of large language models (LLMs) is critical for ensuring they operate ethically, accurately, and responsibly Shen et al. (2023). Proper alignment

---

[1] https://github.com/ollama/ollama

helps prevent the generation of harmful or biased content, promotes factual correctness and reliability, and supports legal compliance and user trust Wang et al. (2023b). Additionally, it ensures that LLMs contribute positively to society by respecting social values, enhancing educational integrity, and maintaining accessibility and inclusivity for all users. As these models become more integrated into daily life, their alignment with societal norms and expectations becomes increasingly important to foster beneficial interactions and outcomes.

Current approaches typically cater to either explicit reward data or preference data, often lacking the versatility to handle both simultaneously. Reinforcement Learning (RL) (Schulman et al., 2017) is well-suited for scenarios with explicit rewards. However, its on-policy nature requires the initial step of learning a reward model from data, resulting in an indirect two-stage optimization process (Christiano et al., 2017; Ouyang et al., 2022; Shen et al., 2024). Recent advancements in preference-based alignment techniques (Rafailov et al., 2024b; Azar et al., 2024; Ethayarajh et al., 2024; Wang et al., 2023a; Hong et al., 2024) have simplified this process, enabling direct alignment of language models (LMs) through a single loss function. Nevertheless, these methods are limited to pairwise preference data. Other alignment strategies (Yuan et al., 2024a; Song et al., 2024) are similarly not designed for use with reward datasets. Recent work like Cai et al. (2023) has attempted to extend DPO's parameterization technique to explicit reward contexts, but it is restricted to binary rewards. In contrast, our methods are capable of handling both continuous rewards and preference data.

## 3 PRELIMINARIES: DIRECT PREFERENCE OPTIMIZATION

LM alignment is essentially a constrained policy optimization problem:

$$\max_{\pi_\theta} \mathbb{E}_{p(x)} \left[ \mathbb{E}_{\pi_\theta(y|x)} r(x, y) - \alpha D_{KL} \left( \pi_\theta(\cdot|x) || \mu(\cdot|x) \right) \right], \tag{1}$$

where $\mu$ represents the pretrained LM. $x$ and $y$ are respectively instructions and responses. $r$ is a reward function that reflects human intentions. $\alpha$ is the temperature coefficient. Peng et al. (2019) has proved that the optimal solution for the optimization problem in Equation 1 is:

$$\pi^*(y|x) = \mu(y|x) \frac{e^{r(x,y)/\alpha}}{Z(x)} \propto \mu(y|x) e^{r(x,y)/\alpha}. \tag{2}$$

Direct Preference Optimization (DPO) (Rafailov et al., 2024b) assumes we only have access to some pairwise preference data $x \to \{y_w > y_l\}$ for each instruction $x$. The preference probability of human annotators is modeled by a learnable implicit reward model $r_\theta$ under Bradley-Terry theories (Bradley & Terry, 1952):

$$\pi_\theta(y_w > y_l|x) := \sigma(r_\theta(y_w, x) - r_\theta(y_l, x)), \tag{3}$$

where $\sigma$ is the sigmoid function. To learn $r_\theta$, DPO simply adopts a binary classification loss:

$$\mathcal{L}_{\text{DPO}} = -\mathbb{E}_{\{x, y_w > y_l\}} \log \sigma \left( r_\theta \left( y_w, x \right) - r_\theta \left( y_l, x \right) \right). \tag{4}$$

In practice, the latent function $r_\theta$ is parameterized by the log-likelihood ratio between $\pi_\theta$ and $\mu$:

$$r_\theta(x, y) := \beta \log \frac{\pi_\theta(y|x)}{\mu(y|x)}, \tag{5}$$

where $\beta$ a linear coefficient for scaling $r_\theta$. This parameterization is crucial because it ensures $\pi^\theta(y|x) \propto \mu(y|x) e^{r_\theta(x,y)/\beta}$ constantly hold. It transforms generative policy optimization into a simple discriminative classification task: When $r_\theta = r$ and $\beta = \alpha$ are satisfied, we naturally have $\pi_\theta = \pi^*$.

## 4 METHOD

### 4.1 PREFERENCE DATA VERSUS REWARD DATA

Compared to constructing preference datasets, annotating each response with scalar rewards can be more flexible and convenient. Preference methods are limited to pairwise comparisons ($x \to$

$\{y_w > y_l\}$) and would require $C_K^2$ evaluations to compare $K$ responses. In contrast, reward datasets $(x \to \{y_i, r_i\}_{1:K})$ allow for an arbitrary number of responses per prompt with just $K$ evaluations.

Despite its simplicity in handling preference data, DPO is not tailored for reward datasets. We introduce a new alignment method termed LPO to mitigate this gap. We show that reward alignment can be solved by constructing a classification problem to identify the optimal response from multiple candidates. We then demonstrate that LPO subsumes DPO as a special case and thus is a natural extension of DPO.

## 4.2 OBJECTIVE

In essence, DPO represents response rewards as LM likelihoods and constructs a pairwise ranking task for learning the reward model. Given that there are more than two ($K > 2$) responses per prompt in reward datasets, we seek to construct a listwise ranking task for learning reward models from explicit rewards instead of preference labels. Note that we want to construct a loss objective that can rank the responses in a pairwise fashion, and injects the reward signals into the ranking process.

Assume that the responses $\{y_i\}_{1:K}$'s annotated rewards $\{r_i\}_{1:K}$ are ranked in a descending order, that is, $r_i > r_j$ if $i < j$. Inspired by the LambdaRank method Burges et al. (2006), we now propose our listwise preference optimization (LPO) method. Formally, the LPO loss is given by:

$$\mathcal{L}_\theta^{LPO} = - \sum_{1 \le i < j \le K} m(i,j) * \log \sigma \left( r_\theta \left( y_i, x \right) - r_\theta \left( y_j, x \right) \right), \tag{6}$$

where $m(i,j)$ is given by

$$m(i,j) = |r_i^2 - r_j^2|, \tag{7}$$

where $| \cdot |$ returns the absolute value. Through the coefficient $m(i,j)$, Equation 6 injects the comparisons among all the responses to the the LLM.

## 4.3 ABSOLUTE VALUE REGULARIZATION

A notable issue with DPO is that the likelihood of all responses consistently decreases throughout the training process (Pal et al., 2024; Rafailov et al., 2024a). We observe that LPO exhibits this same trend due to their inherent equivalence. This reduction in data likelihood is undesirable because it directly contradicts the maximum likelihood objective of supervised training and may ultimately impair performance (Yuan et al., 2024b).

We hypothesize the main cause of this decreasing likelihood is that LPO methods only adjust relative rewards among responses, rather than optimizing their absolute value. To address this problem, we propose LPO-abs, a variant to LPO. LPO-abs can also guarantee convergence to the optimal LM policy by directly learning the absolute reward for each response, thereby counteracting the decreasing likelihood trend. Formally,

$$\mathcal{L}_\theta^{\text{LPO-abs}}(x, \{y_i, r_i\}_{1:K}) = \mathcal{L}_\theta^{\text{LPO}}(x, \{y_i, r_i\}_{i=1}^K) + \frac{1}{K} \sum_{i=1}^K (\log \sigma(-r_\theta(x, y_i))), \tag{8}$$

The loss function for LPO-abs involves two forces that jointly determine the trend of increasing or decreasing $r_\theta(x, y)$. Responses with higher rewards would, in principle, attain higher likelihood after training. Responses with lower rewards will attain proper likelihood under the influence of the two terms.

Table 1 contrasts the optimization objectives of LPO and LPO-abs. Both LPO and LPO-abs adjust the relative values of reward models across various responses $\{y_i\}_1^K$ for a given instruction $x$. In other words, the absolute value of $r_\theta(x, y)$ is not directly constrained, which can lead to counter-intuitive behaviors. For example, the reward for even the highest-ranked response might decrease over time, as long as the reward margin continues to increase, potentially resulting in poor performance or training instability. In contrast, LPO-abs specifically targets the optimization of the absolute values of the reward model. In practice, this approach effectively prevents the likelihood of preferred responses from diminishing. We find this particularly beneficial for challenging tasks such as coding.

Table 1: Comparison of LPO and LPO-abs algorithm for aligning language models. Both reward loss and pairwise preference loss are given.

| Alignment Method | LPO | LPO-abs |
|---|---|---|
| Modeling Target | $\pi^*(y\|x) \propto \mu_t(y\|x)e^{r_t(x,y)/\alpha}$ | |
| Reward dataset | $\mathrm{x} \to \{y_i, r_i\}_{1:K}$ | |
| Loss ($K > 1$) | $-\sum_{1 \le i < j \le K} m(i,j) * \log \sigma\left(r_\theta(y_i, x) - r_\theta(y_j, x)\right)$ | $LPO + \frac{1}{K} \sum_{i=1}^{K} \sigma(-r_\theta(x, y_i))$ |
| Preference Dataset | $x \to \{y_w > y_l\}$ | |
| Loss ($K = 2$) | $-\log \sigma(r_\theta(x, y_a) - r_\theta(x, y_b))$ (DPO) | $DPO + \frac{1}{2} \sum_{i \in \{w,l\}} \sigma(-r_\theta(x, y_i))$ |
| Optimizing Target | relative value | relative and absolute value |
| $\mathrm{r}_{\theta^*}(x, y_{best}) \ge 0$ | not guaranteed | guaranteed |

## 5 EXPERIMENTS

We mainly seek to answer two research questions through our experiments:

- If we have access to reward-annotated datasets with more than 2 responses per prompt, does LPO or LPO-abs offer empirical improvement compared with preference-based approaches that simply prune reward datasets into preference datasets?

- If only pairwise preference data is available, when should one choose LPO-abs over DPO? What benefits does LPO-abs offer? Note that LPO is exactly DPO in this setting.

### 5.1 EXPRIMENTS ON THE REWARD DATASET

**Reward datasets**  We employ the UltraFeedback (Cui et al.) dataset, an instruction-following dataset annotated by GPT-4. UltraFeedback aims at advancing the alignment of large language models (LLMs) with human preferences through the use of large-scale, high-quality, and diversified feedback datasets. Utilizing a vast dataset of approximately 64,000 prompts and 256,000 responses from various high-quality sources, UltraFeedback evaluates LLM responses on criteria such as instruction-following, truthfulness, honesty, and helpfulness. GPT-4 rates each response with a scalar reward on a scale of 0-10. Prior research indicates that these GPT-4 rewards closely align with human annotations (Zheng et al., 2023), establishing them as an efficient, cost-effective alternative to human feedback. This dataset is used to train models to better serve user needs while maintaining ethical standards.

**Evaluation datasets**  We choose the well-acknowledged and widely used GPT4-based benchmarks as follows for evaluating LLMs:

- MT-bench (Zheng et al., 2023). MT-Bench is a benchmark designed to evaluate the performance of chat assistants across multiple turns of conversation and various categories of tasks. It consists of 80 multi-turn questions covering common use cases that are challenging enough to differentiate among models. These questions are categorized into eight types of user prompts, including Writing, Roleplay, Extraction, Reasoning, Math, Coding, STEM, and Humanities/Social Science. Each category includes 10 multi-turn questions crafted carefully by experts. To assess the models' performance on MT-Bench, strong language models (LLMs) are used as judges, and the results indicate that these LLM judges, particularly GPT-4, can align well with both controlled and crowdsourced human preferences, achieving over 80% agreement, which is comparable to the level of agreement between humans. Thus, MT-Bench serves as a comprehensive tool to measure the effectiveness of chat assistants, complementing traditional benchmarks.

- AlpacaEval (Dubois et al., 2024) is an LLM-based automated evaluation metric consisting of a set of 805 instructions that reflect typical user interactions on the Alpaca web demo. Both a baseline model, typically GPT-4 turbo, and the model under evaluation generate responses for each instruction, which are then compared side-by-side by another GPT-4 turbo-based evaluator that outputs the likelihood of preferring the evaluated model's response. The metric calculates a win rate, representing the expected probability that the

evaluated model's output is preferred over the baseline across these instructions. Initially designed for the Alpaca chatbot and AlpacaFarm simulator, AlpacaEval was intended to mitigate certain biases, such as presentation order, by randomizing sequences. However, other factors like length bias were not controlled for, leading to potential manipulation by AI systems. To address this, Length-Controlled AlpacaEval was introduced, which employs a regression-based method to adjust for biases by controlling for length differences, thereby improving the robustness and correlation with human evaluations.

**Evaluation metrics** MT-Bench utilizes the GPT-4 as the judge. As is shown in Zheng et al. (2023), GPT-4 can assign a fair score between 1 to 10 to a LLM's response which is consistent with human annotators. We report the average GPT-4 score (gpt-4-score).

On the AlpacaEval dataset, we compare the LLM against the reference responses from the Davinci-003 (Brown et al., 2020) model. We utilize the GPT-4 as the judge and determine how often the LLM wins the reference responses (denoted as the win rate).

In addition, we will put the LLM aligned with the LPO method or the DPO method in the Chatbot Arena (Chiang et al., 2024), and count the ratio of winning rate against the other models.

**Baselines** We exam the following baseline methods:

- DPO (Rafailov et al., 2024b). To apply the DPO method with reward datasets in which more than two responses are annotated per instruction, we conduct the data transformation according to Zephyr (Tunstall et al., 2023). DPO selects the highest reward response and a random remaining one from UltraFeedback for each instruction. This procedure discards two additional suboptimal responses in the dataset as well as their reward information.

- DPO-1vsO. Comparing the above procedure of constructing DPO dataset, one might predict that applying the DPO to a more grained preference dataset would yield better results. To investigate this, we examined a variants of DPO that utilize all available responses in UltraFeedback by pair the highest-performing response with each of the other ones separately.

- DPO-pw. This variant utilizes all available responses in UltraFeedback by summing up all DPO loss possibilities for two out of all the responses.

- KTO (Ethayarajh et al., 2024). Using a Kahneman-Tversky model of human utility, KTO propose a novel method that directly maximizes the utility of generations instead of maximizing the log-likelihood of preferences, as current methods do.

- IPO (Azar et al., 2024). IPO proposes a new general objective for learning from human preferences that is expressed in terms of pairwise preferences and therefore bypasses both approximations.

**Results** In Table 2, we present the results of fine-tuning a Mistral-7B model on the UltraFeedback dataset and compare our LPO and LPO-abs methods against the baselines. The results demonstrate that our methods outperform recent baseline approaches for preference optimization. This improvement can be attributed to LPO/LPO-abs's ability to fully leverage the information contained within the reward dataset. Among the two versions of our method, LPO-abs outperforms LPO, highlighting the significance of maintaining the absolute values of the best responses' rewards.

## 5.2 EXPRIMENTS ON THE PREFERENCE DATASET

The previous experiments have focused on reward datasets with more than two responses per input prompt ($K > 2$, $x \rightarrow \{y_i, r_i\}_{1:K}$). However, most current alignment datasets are pairwise ($x \rightarrow \{y_w, y_l\}$), necessitating an evaluation of our proposed methods in pairwise preference settings as well. Since LPO is equivalent to DPO when only pairwise preference data is available, we will concentrate on comparing and elucidating the differences between the DPO and LPO-abs algorithms.

**Preferecne dataset** We conduct experiments on the UltraInteract dataset Yuan et al. (2024b). ULTRAINTERACT is a high-quality dataset aimed at enhancing the reasoning abilities of large language models through complex tasks. It supports both supervised fine-tuning and preference learning by providing a preference tree for each instruction, including reasoning chains with diverse

Table 2: Comparison between reward-based methods (SPO, LPO-abs) and preference-based methods (for example, DPO, IPO) in LLM alignment. We focus on the general instruction-following abilities of each method measured by GPT-4 evaluations and human preference. The highest number (excluding the Referenced models ) in each benchmark is **bolded** and the second highest is underlined.

| Method | Annotation Type | MT-bench | AlpacaEval | Win agaist DPO |
|---|---|---|---|---|
| *Referenced models* | | | | |
| GPT-4 | Reward Model | 9.18 | 93.8 | - |
| LLaMA2-chat-70b | Reward Model | 6.86 | 92.7 | - |
| Mistral-7B-instruct | SFT Data | 6.84 | 92.6 | - |
| *Baseline models* | | | | |
| Mixtral-7B-sft | SFT Data | 6.45 | 85.2 | - |
| + KTO | Preference | 7.12 | 91.9 | - |
| + IPO | Preference | 7.45 | 90.6 | - |
| + DPO (Zephyr-$\beta$) | Preference | 7.34 | 90.6 | - |
| + DPO-1vsO | Preference | 7.22 | 91.6 | 52.1 |
| + DPO-pw | Preference | 7.38 | 90.3 | 53.3 |
| *Ours* | | | | |
| + LPO | Reward | 7.57 | 91.3 | 54.7 |
| + LPO-abs | Reward | **7.63** | **92.0** | **57.1** |

strategies, multi-turn interactions, and paired correct and incorrect data. The dataset was curated with an emphasis on complexity, quality, and diversity, focusing on math problem-solving, code generation, and logical reasoning. It includes ground-truth solutions to ensure high-quality oversight signals. We consider fine-tuning Mistral-7B and Mistral-8×7B models on UltraInteract. Before alignment, we perform SFT on UltraInteract's preferred responses.

**Evaluation datasets** We evaluate the model's performance in various challenging tasks. This includes:

- BBH (Suzgun et al., 2022). It is a subset of the BIG-Bench, a diverse evaluation suite for language models. BBH focuses on a suite of 23 challenging tasks from BIG-Bench that were found to be beyond the capabilities of current language models. These tasks are ones where prior language model evaluations did not outperform the average human-rater. The BBH tasks require multi-step reasoning, and it was found that few-shot prompting without Chain-of-Thought (CoT), as done in the BIG-Bench evaluations, substantially underestimates the best performance and capabilities of language models. When CoT prompting was applied to BBH tasks, it enabled PaLM to surpass the average human-rater performance on 10 of the 23 tasks, and Codex to surpass the average human-rater performance on 17 of the 23 tasks.

- HumanEval Chen et al. (2021). It used to measure functional correctness for synthesizing programs from docstrings. It consists of 164 original programming problems, assessing language comprehension, algorithms, and simple mathematics, with some comparable to simple software interview questions.

- LeetCode Guo et al. (2024). LeetCode presents competition-level problems, offering significant challenges that test the model's problem understanding and code generation skills.

- Math tasks. In this work, we include 5 challenging tasks for math problem solving: (a) GSM-Plus Li et al. (2024). (b) MATH (Hendrycks et al., 2021). (c) TheoremQA (Chen et al., 2023). (d) SVAMP (Naeem et al., 2014). (e) ASDiv (Miao et al., 2021).

**Evaluation metrics**

**Results** Results are presented in Table 3. Overall, LPO-abs consistently outperforms DPO across various benchmarks. Notably, we observe that DPO sometimes hampers overall performance in certain reasoning tasks compared to the SFT models. This suggests that DPO may not be well-suited for enhancing reasoning abilities, a finding that aligns with concurrent research (Yuan et al., 2024b). In contrast, LPO-abs demonstrates clear improvements on the 7B models.

Table 3: Comparison between LPO-abs and DPO in LLM alignment. We focus on the reasoning capabilities on the complex tasks. The highest number in each benchmark is **bolded**. We mark numbers that have decreased (↓) after training.

|  | Model | Mixtral-7B | | |
|---|---|---|---|---|
|  |  | SFT | + DPO | + LPO-abs |
| Reasoning | BBH | 60.8 | 61.1 | **61.4** |
| Coding | HumanEval | 27.8 | **30.8** | 30.3 |
|  | LeetCode | **3.3** | 2.4 (↓) | 3.3 |
| Math | GSM-Plus | 28.3 | 19.4 (↓) | 30.2 |
|  | MATH | 5.8 | 6.4 | **9.7** |
|  | TheoremQA | 7.1 | **8.9** | 8.7 |
|  | SVAMP | 26.9 | 34.1 | **41.5** |
|  | ASDiv | 40.8 | 46.1 | **50.3** |

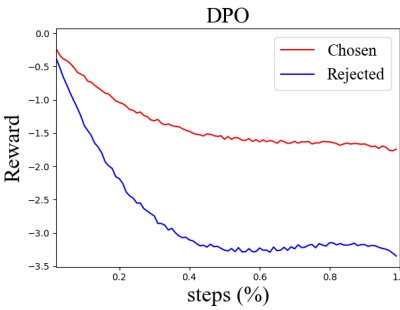 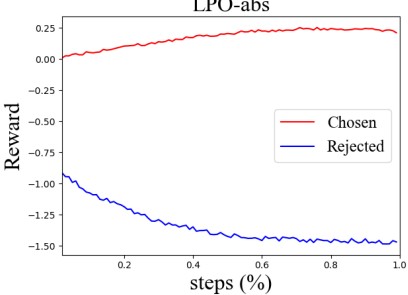

Figure 1: Comparision of data likelihood between DPO and LPO-abs, on the preference dataset UltraInteract.

## 5.3 FURTHER ANALYSIS

**Visualization of likelihood for the chosen and rejected** To illustrate the distinct optimization characteristics that lead to performance differences between LPO-abs and DPO, we examine how the data likelihood evolves during training. The results are presented in Figure 1. As shown in Figure 1, the likelihood of preferred responses intriguingly decreases after DPO training, whereas it increases with LPO-abs training. The trend of decreasing chosen likelihood is concerning because it directly contradicts the maximum-likelihood objective utilized during the SFT stage. This drawback is particularly pronounced in reasoning tasks, where the preferred response is often the ground truth answer. Consequently, we hypothesize that LPO-abs exhibits superior performance in reasoning tasks due to its ability to avoid reducing the likelihood of chosen responses. Since DPO can be considered a specialization of LPO, the contrasting likelihood trends can be theoretically explained. Specifically, LPO-abs adjusts the absolute likelihood of data, whereas DPO and LPO focus only on the relative likelihood across different responses. Therefore, a declining chosen likelihood directly conflicts with LPO-abs's training objective but not with DPO's.

**Effects of the number of responses** $K$ Figure 2 plots how different values of $K$ affects the LLM alignment performance. From Figure 2, we observe consistent performance improvements when increasing the number of data responses from $K = 2$ to $K = 4$ for both LPO and LPO-abs algorithms. In addition, we observe the LPO-abs outperforms LPO under different settings, demonstrating the importance of maintaining the absolute values for the best response's reward.

Figure 2 also conveys an important take-away. Previous practices always ensure selecting the highest performing response when constructing preference data. The assumption behind this strategy is that the dataset's best-performing response determines the upper limit of alignment performance.

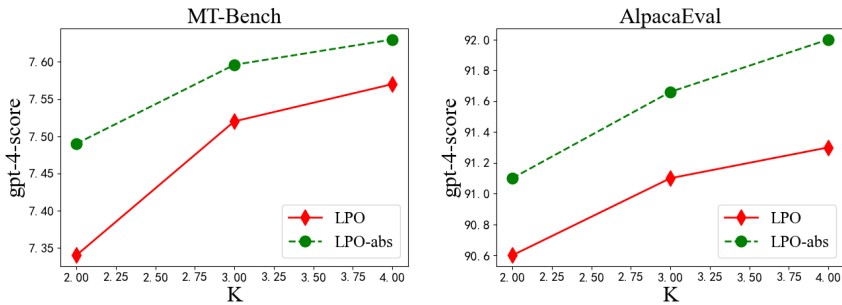

Figure 2: Evaluation results under different numbers of responses.

However, our experiments contradict this assumption. Results in Figure 2 indicate that extra sub-optimal responses can also be advantageous for policy training.

## 6 CONCLUSION

In this work, we conduct a thorough investigation into the language model alignment problem within the context of explicit reward settings. Drawing inspiration from listwise rank learning, we introduce two novel and practical algorithms: LPO and LPO-abs. Our proposed methods are uniquely suited for both reward data and preference data, encompassing DPO as a special case. Our experiments demonstrate that our reward-based alignment methods, LPO and LPO-abs, outperform preference-based baselines such as DPO and KTO by fully leveraging sub-optimal responses in reward datasets. In preference settings, the pairwise version of LPO-abs surpasses DPO in complex reasoning tasks by effectively preventing the likelihood of preferred responses from decreasing.

## ETHICS STATEMENT

The authors are committed to upholding the highest standards of ethical research practice. This study did not involve direct interaction with human subjects; however, we acknowledge the importance of addressing potential ethical concerns related to our work. The datasets utilized in this research were obtained from publicly available sources and do not contain any personally identifiable information. To ensure the responsible use of these datasets, we adhered to all relevant guidelines concerning data privacy and security.

Our methodology was designed with the aim of promoting fairness and reducing bias. We acknowledge that machine learning models can inadvertently propagate biases present in training data. Therefore, we have taken steps to mitigate such risks by employing techniques to detect and correct for biases within our dataset.

Additionally, the authors declare no conflicts of interest or financial ties that could influence the results or interpretation of this study. Any sponsorship received for this project has been noted and has not influenced the research outcomes.

This work complies with all relevant institutional and governmental regulations concerning research involving computational models and artificial intelligence. The research was conducted in accordance with the principles outlined by institutional review boards (IRBs) and other governing bodies as applicable. Documentation supporting these statements is available upon request from the corresponding author.

We are aware of the potential implications of our findings and emphasize the need for caution when applying these methodologies in real-world scenarios. It is important that any application of our findings be done responsibly, taking into account the broader societal impact.

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
