# OpenReview forum: "A Novel Listwise Alignment Approach for Language Models with Explicit Rewards"
_ICLR.cc/2025/Conference — ICLR 2025 Conference Withdrawn Submission_

### Official Review · Reviewer_G9Sb · 2024-10-31

**Soundness:** 2
**Presentation:** 3
**Contribution:** 1
**Rating:** 3
**Confidence:** 5

**Summary:**

The paper describes a way to to policy optimization when mutliple generations for a single prompt are available together with their true reward scores. They authors then proposed to use a weighted DPO objective based on the difference of the given reward scores. They in addition also propose a modification to enhance the likelihood decay. They show on experiments that their proposed method does somewhat better that existing and naive DPO.

**Strengths:**

- The paper explores a different preference/reward model dataset where rewards are given and used to weigh the objective
- The paper is well-written and easy to follow
- The paper is straightforward and easy to implement if data is available.

**Weaknesses:**

- The paper's contribution is minimal as they merely add a weight based on the reward difference. The motivation is unclear and seems to be the most obvious way to include rewards when they are present. The idea of weighting the DPO loss is not novel and simply using the difference of rewards, in my opinion, does not warrant a paper unless the results are exceptional. [1] Please let me know if i have missed something.
- The objective LPO-abs doesn't seem to make sense to me. How does increasing the reward on every sample help the model at all? Should this be the difference between the true reward and the predicted reward? i.e., eq (9). Please explain the regularization term to me again, as I might have misunderstood.
- The paper also crucially fails to compare against RPO i.e. SFT+DPO [2] Which also solves the likelihood problem raised by the authors. However, no comparisons were made and I suspect that it will do similarly if not almost exactly the same as LPO-abs in the pairwise case. Please let me know why this wasn't added, happy to be proven wrong.
- I am also extremely surprised that there is no hyperparameter for the regularization term in LPO-abs. Is this a feature of this method? if yes please explain the intuition behind this property.




[1] WPO: Enhancing RLHF with Weighted Preference Optimization
[2] Provably mitigating overoptimization in rlhf: Your sft loss is implicitly an adversarial regularizer

**Questions:**

I have raised my concerns above please address them.

---

### Official Review · Reviewer_PGxV · 2024-10-31

**Soundness:** 3
**Presentation:** 3
**Contribution:** 3
**Rating:** 5
**Confidence:** 4

**Summary:**

The paper proposes LPO for pointwise reward datasets, which generalizes DPO by assigning weights to different pairs of responses. It also proposes to fix the data likelihood decline issue with an extra cross entropy term. The experiments show improved performance across tasks and datasets.

**Strengths:**

* It studies pointwise reward data setting for offline alignment methods. This is an important problem and seems under-explored.
* it studies the data likelihood decline issue, which is known to cause problems.
* It conducts extensive experiments.

**Weaknesses:**

* There seems to be a confusing typo --- in the abstract the method is called Soft Preference Optimization?
* I am not completely convinced by the design of the LPO loss. While it is a sensible generalization of the DPO loss (weighting pairs with bigger reward differences), it's not clear if this is the most principled choice. The derivation of DPO itself, relies on Bradley Terry assumption  on the preference data. I feel a better derivation of the loss might come from statistical assumptions on how to model the pointwise data: e.g. does bigger pointwise reward difference mean bigger preference difference, or bigger certainty in the preference? The current formulation seems to use pointwise reward difference as a proxy for uncertainty (more certain pairs gets more weights). But is it a good choice? There are certainly many alternatives that the authors can study/compare against.
* About data likelihood decline issue: other papers (e.g. [1]) use the cross entropy of the preferred samples as the extra term, and seem to get rid of the issue. The authors can compare against that.

[1] The Llama 3 Herd of Models https://arxiv.org/abs/2407.21783

**Questions:**

I wrote my questions in the weakness part.

---

### Official Review · Reviewer_Sy9u · 2024-11-02

**Soundness:** 2
**Presentation:** 3
**Contribution:** 2
**Rating:** 3
**Confidence:** 3

**Summary:**

This paper presents LPO (Listwise Preference Optimization), a method for training language models using reward datasets. The authors propose LPO and its variant LPO-abs, which they claim can optimize language models using reward data and address the data likelihood decline problem observed in DPO. The authors describe their methods as being capable of working with both reward data and preference data in a unified framework. Through experimental evaluation, the authors report that LPO outperforms existing preference-based methods when tested on both reward and preference datasets, which they attribute to better utilization of information from reward datasets. The work aims to contribute to language model training by offering a framework that integrates both reward and preference-based optimization approaches.

**Strengths:**

1. The paper has a clear structure in presenting the methodology. The experimental design follows a good practice, with research questions outlined before results. This helps readers follow the technical contributions despite the complexity of the subject matter.

2. This paper raises an interesting direction to utilize the explicit reward data when preference optimization. The authors consider both reward-based and preference-based scenarios in their experiments even though their method is mainly about reward-based scerario. This is necessary since in general, the explicit reward dataset is not well defined and often hard to get, which makes the preference datasets still the dominant real applicational case.

3. The evaluation covers a good range of benchmarks across different domains (BBH, HumanEval, LeetCode, various math tasks). This testing across diverse tasks helps understand how those methods compares in different domains.

**Weaknesses:**

1. The reward-based experiments in Section 5.1 raise methodological concerns regarding evaluation bias. Specifically, the reported improvements on MT-bench warrant careful interpretation given that the model's training utilizes GPT-4 rewards (0-10 scale) and is subsequently evaluated using the same GPT-4 scoring system. This circular evaluation framework makes it challenging to disentangle genuine methodological improvements from potential artifacts introduced by the training data. A more robust evaluation would benefit from independent metrics and human evaluation to validate the reported gains.

2. The AlpacaEval results are actually comparable to baselines, with some labeling inconsistencies in reporting the second-best results; Moreover, the absence of win rate (WR) reporting against DPO for some baseline,(e.g. KTO, the best baseline in AlpacaEval metric), limits our ability to fully assess the method's comparative advantages.

3. Although the explicit reward data setting is worth exploration, the technical novelty of the method is limited from my perspective. The core idea essentially multiplies the DPO objective by the absolute difference of reward values between winning and losing samples, then adds a term to keep the learned reward generally high. This modification is relatively straightforward and the empirical results don't demonstrate significant improvements to justify these design choices.

4. The loss function design appears largely heuristic. While drawing inspiration from LambdaRank, there's insufficient theoretical justification for why this particular formulation is optimal. Given the lack of strong theoretical grounding, the empirical results would need to be more convincing to justify the design choices.

**Questions:**

1. Using explicit reward data for LLM alignment is an interesting direction that could potentially offer more direct supervision than pairwise preferences. However, this paradigm raises several questions that deserve deeper discussion: How should we define "good" explicit rewards for language model alignment? While this paper uses GPT-4 scores, we should consider more broadly: what are principled ways to collect such reward data? Different sources (human annotations, model scores, automated metrics) might introduce different biases - for example, using GPT-4 scores might bias the model toward GPT-4's behavior rather than true human preferences. A deeper discussion of these considerations would help establish best practices for using explicit rewards in alignment.

2. The paper introduces a method to avoid the reward decreasing of all responses $r\theta(x, y_i)$ in LPO-abs, but the theoretical foundation could be further developed. Since standard reinforcement learning theory suggests that policies remain unchanged when rewards are shifted by a constant, it would be valuable to better understand why maintaining absolute reward values is beneficial in this context. If so, then why is it necessary a bad thing that chosen response’s reward decreases as long as the difference between wining and losing samples is learne? Additional theoretical analysis exploring this aspect would make the motivation clearer and could reveal interesting insights about language model training dynamics.

---

### Official Review · Reviewer_92iv · 2024-11-04

**Soundness:** 3
**Presentation:** 3
**Contribution:** 3
**Rating:** 5
**Confidence:** 3

**Summary:**

This paper introduces Listwise Preference Optimization (LPO) which processes listwise preference information, leveraging a novel optimization objective tailored for multiple responses per prompt. This listwise approach supports both reward and preference datasets, incorporating DPO as a special case. The paper also introduces LPO-abs, a variant designed to counteract the issue that response likelihoods decrease over training. Experimental results across challenging tasks, including mathematical reasoning and coding, demonstrate that LPO and LPO-abs outperform baseline models like DPO and KTO.

**Strengths:**

It proposes an innovative method modified from DPO for listwise alignment, which is novel.

The evaluation datasets are comprehensive and diverse.

The proposed LPO-abs effectively prevent the likelihood of preferred responses from decreasing, which solve an important issue of DPO.

**Weaknesses:**

While LPO and LPO-abs are evaluated across several benchmarks, they are only experimented on the mistrial language model.

The baseline methods are not comprehensive enough.

Also, the improvement seems to be slight right now.

**Questions:**

Please refer to the weaknesses.
1. To validate the robustness of the alignment method, other models like Llama may also be tested.
2. The baseline methods may include others like IPO, SimPO, etc to fully validate the performance.
3. It would be better to compare the method with some other reward-based alignment method that can work on the reward datasets

---

### Public Comment · ~Daniil_Gavrilov1 · 2024-11-13
**Double submission**

This paper is a duplicate of the submission at https://openreview.net/forum?id=28TLorTMnP (see Table 2, 3, Figure 1, 2). The version with the lower scores was withdrawn.

---

### Note · Authors · 2024-11-13

I have read and agree with the venue's withdrawal policy on behalf of myself and my co-authors.